# Insight into the Interplay of Gd-IgA1, HMGB1, RAGE and PCDH1 in IgA Vasculitis (IgAV)

**DOI:** 10.3390/ijms25084383

**Published:** 2024-04-16

**Authors:** Martina Held, Ana Kozmar, Mario Sestan, Daniel Turudic, Nastasia Kifer, Sasa Srsen, Alenka Gagro, Marijan Frkovic, Marija Jelusic

**Affiliations:** 1Department of Pediatrics, University of Zagreb School of Medicine, University Hospital Centre Zagreb, 10000 Zagreb, Croatia; martina.held@mef.hr (M.H.); mario.sestan@gmail.com (M.S.); danielturudic@gmail.com (D.T.); nastasia.ce@gmail.com (N.K.); mfrkovic1@gmail.com (M.F.); 2Department of Laboratory Diagnostics, University of Zagreb School of Medicine, University Hospital Centre Zagreb, 10000 Zagreb, Croatia; ana.kozmar@kbc-zagreb.hr; 3Department of Pediatrics, University of Split School of Medicine, University Hospital Centre Split, 21000 Split, Croatia; ssrsen@kbsplit.hr; 4Children’s Hospital Zagreb, Medical Faculty Osijek, Josip Juraj Strossmayer University of Osijek, 10000 Zagreb, Croatia; alenka.gagro@gmail.com

**Keywords:** IgA vasculitis, pathogenesis, biomarkers

## Abstract

The pathogenesis of IgAV, the most common systemic vasculitis in childhood, appears to be complex and requires further elucidation. We aimed to investigate the potential role of galactose-deficient immunoglobulin A1 (Gd-IgA1), high-mobility group box 1 (HMGB1), receptor for advanced glycation end products (RAGE) and protocadherin 1 (PCDH1) in the pathogenesis of IgAV. Our prospective study enrolled 86 patients with IgAV and 70 controls. HMGB1, RAGE, Gd-IgA1 and PCDH1 in serum and urine were determined by the enzyme-linked immunosorbent assay (ELISA) method at the onset of the disease and after a six-month interval in patients and once in the control group. Serum concentrations of HMGB1, RAGE and PCDH1 and urinary concentrations of HMGB1, RAGE, Gd-IgA1 and PCDH1 were significantly higher in patients with IgAV than in the control group (*p* < 0.001). Concentrations of HMGB1 (5573 pg/mL vs. 3477 pg/mL vs. 1088 pg/mL, *p* < 0.001) and RAGE (309 pg/mL vs. 302.4 pg/mL vs. 201.3 pg/mL, *p* = 0.012) in the serum of patients remained significantly elevated when the disease onset was compared with the six-month follow-up interval, and thus could be a potential marker of disease activity. Urinary concentration of HMGB1 measured in the follow-up period was higher in patients with nephritis compared to IgAV without nephritis (270.9 (146.7–542.7) ng/mmol vs. 133.2 (85.9–318.6) ng/mmol, *p* = 0.049) and significantly positively correlated with the urine albumine to creatinine ratio (τ = 0.184, *p* < 0.05), the number of erythrocytes in urine samples (τ = 0.193, *p* < 0.05) and with the outcome of nephritis (τ = 0.287, *p* < 0.05); therefore, HMGB1 could be a potential tool for monitoring patients with IgAV who develop nephritis. Taken together, our results imply a possible interplay of Gd-IgA1, HMGB1, RAGE and PCDH1 in the development of IgAV. The identification of sensitive biomarkers in IgAV may provide disease prevention and future therapeutics.

## 1. Introduction

IgA vasculitis (IgAV), previously described by the eponym Henoch–Schönlein purpura (HSP), is a non-granulomatous necrotizing inflammation of small blood vessels mainly containing deposits of IgA immune complexes [1,2,3]. It is characterized by skin changes in the form of nonthrombocytopenic palpable purpura or petechiae usually affecting the lower limbs followed by joint, gastrointestinal and renal involvement [1,2]. IgAV is predominantly a childhood disease, with an estimated annual incidence of 3–55.9 cases per 100,000 children, while the incidence in adulthood is lower and ranges from 0.8–1.8 per 100,000 adults [1,4,5,6]. In children, IgAV is generally a self-limiting disease with favorable resolution; however, a significant percentage of patients develop IgA vasculitis nephritis (IgAVN) within the clinical course. It represents a major long-term complication of IgAV that can progress to end-stage renal disease (ESRD) in 1–5% of children, while this percentage in adult patients can reach up to 40% [7,8,9].

The current evidence indicates that IgAV occurs in genetically susceptible individuals primarily as a response to various environmental factors, particularly infections and chemical agents. The characteristic vascular deposition of IgA1 suggests that in IgAV, there is predominantly an IgA-mediated dysregulated immune response to antigens, forming immune complexes that further activate the alternative and mannan-binding lectin complement pathways and recruit immune cells, leading to endothelial injury in the skin, synovial membrane, gut and kidneys [10,11].

Although considered an immune complex-mediated disease, the pathogenesis of IgAV appears to be more complex, with many questions remaining unanswered, and further studies are necessary to fully understand it. Current perspectives are directed towards promising biomarker discoveries. In the context of IgAV, biomarkers may have clinical utility in multiple non-mutually exclusive targets, including susceptibility to the disease, understanding pathogenesis, assessing disease activity, predicting outcomes—particularly nephritis, the most significant chronic complication of the disease—and guiding therapeutic choices [12].

Galactose-deficient IgA1 (Gd-IgA1) is the most extensively studied potential biomarker in patients with IgAV, and it is generally accepted that Gd-IgA1 plays a role in the pathogenesis of nephritis in IgAV [13,14,15,16,17,18]. In the majority of reported studies, serum levels of Gd-IgA1 were significantly higher in both children and adult patients with nephritis compared to IgAV patients without nephritis [13,14,15,16,17,18,19,20,21]. Two studies indicated that Gd-IgA1 is exclusively present in patients with IgAV nephritis, while there was no difference in serum concentrations between IgAV patients without nephritis and healthy individuals [16,22]. Nevertheless, the role of Gd-IgA1 in IgAV remains controversial, and it is still unknown whether Gd-IgA1 is involved in the pathogenesis of systemic inflammation as well.

High-mobility group protein box 1 (HMGB1) is a ubiquitous single-chain nuclear protein that, when released into the extracellular space, acts as an alarmin and mediates tissue damage. One meta-analysis reported that serum concentrations of HMGB1 were significantly elevated in patients with autoimmune diseases and may reflect disease activity [23]. Although HMGB1, as a proinflammatory cytokine, is implicated in the pathogenesis of various inflammatory and chronic diseases such as rheumatoid arthritis, juvenile idiopathic arthritis, systemic lupus erythematosus, ANCA-associated vasculitides, tumors, cardiovascular diseases and atherosclerosis, only a few studies have investigated its role in IgAV [23,24,25,26,27,28,29,30,31].

Receptor for advanced glycation end products (RAGE) is a transmembrane protein from the immunoglobulin superfamily with a signal transmission function that can bind advanced glycation end products (AGEs) and exists in vivo as a multiligand receptor and in a soluble form (sRAGE) [32]. Transmembrane RAGE on monocytes, macrophages and other cells is overexpressed in response to an increasing concentration of its principal ligand HMGB1, triggering an inflammatory immune response. On the other hand, although lacking transmembrane and intracellular segments, sRAGE acts as a decoy receptor by competitively binding RAGE ligands, thereby antagonizing RAGE-mediated proinflammatory effects [24,27,32]. Previous research has documented the role of RAGE in various chronic and inflammatory diseases such as autoimmune diseases, malignant diseases, rheumatoid arthritis, diabetes, aging processes and cell death [24,25,27,32,33]. While it has also been recognized in systemic vasculitides, the role of RAGE in IgAV has not been described [29,34,35].

Protocadherin-1 (PCDH1) is a calcium-dependent transmembrane glycoprotein present in a variety of cells and tissues, including keratinocytes, fibroblasts, endothelial cells, peripheral mononuclear cells, respiratory epithelium and kidneys [36]. Dysfunction of PCDH1 has been described in diseases such as eczema, atopic dermatitis, asthma and in some malignant diseases, where the presumed mechanism of disease development is the loss of intracellular adhesion and incomplete differentiation of the epithelial layer, contributing to a weaker protective function [37,38,39,40,41,42]. PCDH1 has not been investigated in autoimmune rheumatic diseases so far, including IgAV.

The objective of this study was to evaluate Gd-IgA1, HMGB1, RAGE and PCDH1 as potential biomarkers in children with IgAV. We hypothesized that these molecules are involved in the pathogenesis of IgAV. Therefore, we compared concentrations of Gd-IgA1, HMGB1, RAGE and PCDH1 between patients and controls. Additionally, we sought to analyze the association of the obtained results with the disease activity score, as well as other clinical, biochemical and immunological parameters in patients with IgAV.

## 2. Results

### 2.1. Baseline Characteristics of IgAV Patients

Out of 86 patients with IgAV, there were 49 girls and 37 boys with a median age of 6.4 (IQR 4.5–7.8) years (minimum age: 2.25 years and maximum age: 12.75 years).

All patients exhibited skin changes, 76 patients (88.4%) had experienced musculoskeletal system involvement in the form of arthritis and/or arthralgias, 39 patients (45.3%) had gastrointestinal (GI) system involvement, while 26 patients (30.2%) developed nephritis during the clinical course of the disease. Scrotal involvement was detected in six boys (16.2%).

The most commonly used drugs were non-steroidal anti-inflammatory drugs (NSAIDs) and glucocorticoids.

At least one recurrence of IgAV was recorded in 21 patients (24.4%).

The median PVAS assessed at the onset of the disease was 4 (IQR 2–6) with a minimum score of 1 and a maximum score of 12.

Demographic data and clinical features of patients with IgAV are presented in Table 1.

### 2.2. Laboratory Parameters

The comparison of laboratory findings in patients with IgAV at two intervals (at the onset of the disease and after six months of follow-up) and the control group is summarized in Table 2.

The IgAV group at the onset of the disease had significantly higher levels of ESR, CRP, leukocyte count, neutrophils, platelets, fibrinogen, D-dimer, uACR and IgA compared to the control group, and lower values of hemoglobin, serum creatinine, total proteins and serum albumins in comparison with the controls.

Moreover, significant differences in some laboratory parameters were observed between the two groups even when comparing the follow-up period of IgAV patients with that of the controls. At the onset of the disease, patients had significantly higher levels of ESR, CRP, leukocyte and neutrophil count, serum urea, fibrinogen, D-dimer test, ferritin, IgA, IgG, complement components C_3_ and C_4_, total complement activity and fecal calprotectin, and patients were more frequently positive for FOBT, while erythrocyte count, aPTT, total proteins and serum albumin were significantly lower compared to the six-month follow-up period.

### 2.3. Serum and Urine Gd-IgA1, HMGB1, RAGE and PCDH1 in Patients with IgAV and Controls

Concentrations of HMGB1 [5573 (2274–13,829) vs. 1088 (574.3–2942) pg/mL, *p* < 0.001], RAGE [309.9 (166.6–432.1) vs. 201.3 (112.7–319.6) pg/mL, *p* = 0.005] and PCDH1 [44.5 (28.2–61.5) vs. 18.8 (1.2–45.4) ng/mL, *p* < 0.001] in serum, and Gd-IgA1 [10.8 (6.3–21.2) vs. 10.8 (6.3–21.2) mg/mmol, *p* < 0.001], HMGB1 [178.4 (96.7–596.9) vs. 57.1 (36,9–168) ng/mmol, *p* < 0.001], RAGE [12.2 (7.3–21.2) vs. 5.9 (4.1–10.4) ng/mmol, *p* < 0.001] and PCDH1 [174.6 (72.8–327.5) vs. 71.1 (2.1–138.1) ng/mmol, *p* < 0.001] in urine, measured at the onset of the disease, were significantly higher in patients with IgAV than in the control group (Table 3).

Serum and urine concentrations of the same molecules remained significantly higher compared to the controls even after a six-month follow-up interval.

Serum concentration of HMGB1 and RAGE at the onset of the disease was statistically significantly higher compared to the follow-up period, thus possibly indicating an active disease (Figure 1).

### 2.4. Serum and Urine Gd-IgA1, HMGB1, RAGE and PCDH1 in Patients with IgAV without Nephritis and with Nephritis

Concentrations of Gd-IgA1, HMGB1, RAGE and PCDH1 in serum and urine did not differ statistically between patients with nephritis and patients without nephritis at the onset of the disease.

Urinary HMGB1 was significantly higher in IgAV patients with nephritis compared to those without during the follow-up interval (Table 4).

### 2.5. ROC Analysis

Receiver operating characteristic (ROC) analysis was performed to define the optimal cut-off value for distinguishing between the patient and control group. The area under the ROC curve (AUC) determined the cut-off value. A significant AUC was determined for serum concentrations of HMGB1, RAGE and PCDH1, as well as for urine concentrations of Gd-IgA1, HMGB1, RAGE and PCDH1 in discriminating between the patient group and the control group with associated sensitivity and specificity (Table 5).

### 2.6. Logistic Regression Analysis

Univariate logistic regression analysis showed that serum HMGB1 and serum and urinary RAGE are predictors of IgAV after model adjustment (Table 6).

Regarding different clinical manifestations, logistic regression identified that serum concentrations of Gd-IgA1 (CI 0.943–0.992, *p* = 0.028), RAGE (CI 0.983–0.998, *p* = 0.026) and PCDH1 (CI 1.021–1.137, *p* = 0.012), and urinary HMGB1 (CI 1.000–1.002, *p* = 0.026) are predictors of arthritis in IgAV.

Logistic regression did not show any of the investigated molecules as significant predictors of nephritis in IgAV.

### 2.7. Correlation of Gd-IgA1, HMGB1, RAGE and PCDH1 with Clinical Characteristics and Disease Activity

In patients with IgAV, serum RAGE concentrations showed a significant positive correlation with musculoskeletal manifestations, arthritis and arthralgias, respectively (τ = 0.185, *p* = 0.030). Furthermore, a significant positive correlation was found between serum and urine PCDH1 concentrations and arthritis, respectively (τ = 0.182, *p* < 0.05; τ = 0.238, *p* = 0.001).

There were no statistically significant correlations of serum and urinary Gd-IgA1, HMGB1, RAGE and PCDH1 concentrations with PVAS.

### 2.8. Correlation of Gd-IgA1, HMGB1, RAGE and PCDH1 with Laboratory Parameters in IgAV Patients at the Onset of the Disease and after Six-Month Follow-Up

At the disease onset, a significant positive correlation was found in serum concentrations of HMGB1 with CRP (τ = 0.161, *p* = 0.029), ferritin (τ = 0.219, *p* = 0.004) and IgG (τ = 0.147, *p* = 0.045); in serum concentrations of RAGE with IgG (τ = 0.145, *p* = 0.048); and between serum concentrations of PCDH1 and platelets (τ = 0.218, *p* = 0.003). After a six-month follow-up, serum concentrations of HMGB1 and RAGE positively correlated with 24 h proteinuria values (τ = 0.199, *p* < 0.05; τ = 0.152, *p* < 0.05), while serum concentrations of Gd-IgA1 were significant positively correlated with serum urea (τ = 0.147, *p* < 0.05) and eGFR (τ = 0.183, *p* < 0.05), and negatively correlated with 24 h proteinuria values (τ = −0.218, *p* < 0.05). There was still a significant positive correlation between serum PCDH1 and platelet counts (τ = 0.242, *p* < 0.05).

Urinary Gd-IgA1, HMGB1, RAGE and PCDH1 showed a significant positive correlation with uACR in the acute phase of the disease, and this correlation remained significantly positive in the follow-up interval for urinary HMGB1, RAGE and PCDH1, respectively (τ = 0.184, *p* < 0.05; τ = 0.298, *p* < 0.05; τ = 0.184, *p* < 0.05). Urine concentrations of HMGB1 and RAGE showed a significant positive correlation with eGFR as well (τ = 0.254, *p* < 0.05; τ = 0.182, *p* < 0.05).

Additionally, a significant positive correlation was found between urine concentrations of HMGB1 and erythrocytes in the urine test at the onset of the disease (τ = 0.183, *p* < 0.05) and after the follow-up interval (τ = 0.193, *p* < 0.05).

## 3. Discussion

Although it has been recognized for over 200 years [43] and with well-established epidemiology and clinical features, the pathogenesis of IgAV remains unclear. Shedding light on the exact pathomechanisms may enable disease prevention, particularly of severe forms, and provide new therapeutic approaches.

Recently, it has been suggested that IgAV can be viewed as a dual disease explaining both systemic and renal manifestations of the disease [10]. It is hypothesized that galactose-deficient IgA1 (Gd-IgA1) plays a role in the pathogenesis of IgAV, especially in patients who develop nephritis. Aberrantly glycosylated IgA1 acts as an autoantigen to the host and is recognized by antiglycan antibodies. IgG and IgA antibodies form polymeric circulating immune complexes with Gd-IgA1 (Gd-IgA1-IgA, Gd-IgA1-IgG) and, additionally, Gd-IgA1 forms immune complexes with its soluble CD89 receptor (Gd-IgA1-sCD89). These immune complexes are too large to access the space of Disse in the liver and avoid degradation by hepatic cells, and therefore accumulate in the blood. Finally, Gd-IgA1-containing immune complexes deposit in the tissue, probably in the mesangium, where they bind to the transferrin receptor CD71 and trigger an inflammatory cascade, resulting in renal injury in IgAV [10,11,44]. Although this model explains the pathogenesis of nephritis, the role of Gd-IgA1 complexes is not elucidated in IgAV without nephritis, so an alternative hypothesis involving anti-endothelial cell antibodies (AECA) has been proposed. It is assumed that after infection with microorganisms that have similar antigenic structures as human vessel walls, AECA cross-react on their own small vessels. Increased serum levels of IgA1-AECA bind to β2-glycoprotein I (β2-GPI) on endothelial cells, activating the MEK/REK signaling pathway [45,46,47,48,49]. Elevated proinflammatory levels of TNF-α promote the expression of endothelium-related adhesion molecules ICAM-1 (intercellular adhesion molecule-1), VCAM-1 (vascular cell adhesion molecule-1), E-selectin, and L-selectin [50,51]. Activated endothelial cells release from their Weibel–Palade bodies P-selectin and IL-8, potent chemoattractants for neutrophils and granulocytes. Furthermore, the interaction between IgA1 and FcαRI leads to the release of leukotriene B4 (LTB4), thus forming a positive feedback loop of further neutrophil migration [49,52]. Activated neutrophils secrete their enzymes (myeloperoxidases, elastases, proteases, cathepsin G), produce reactive oxygen species (ROS), form neutrophil extracellular traps (NETs) and cause endothelial damage by the process of antibody-dependent cellular cytotoxicity (ADCC) [10,11,53]. The disintegration and necrosis of neutrophils with the scattering of their nuclei, extravasation of erythrocytes, accumulation of fibrin and small blood vessel permeability and edema are the hallmark histologic features illustrating leukocytoclastic vasculitis in IgAV.

The present study supports the involvement of traditional inflammatory molecules such as CRP and ESR, immune system components such as complement activation and immunoglobulin secretion in the pathogenesis of IgAV, as patients with IgAV and control subjects significantly differed regarding these parameters, as expected. Additionally, four molecules were measured in patients with IgAV with the aim of investigating their role and interplay in this disease.

Furthermore, this study has shown that patients with IgAV had significantly higher serum concentrations of HMGB1, RAGE and PCDH1 and urine concentrations of Gd-IgA1, HMGB1, RAGE and PCDH1 than control subjects. Among them, serum HMGB1 and serum and urine RAGE are distinguished as predictors of IgAV after regression analysis, implying their possible role in the development of IgAV.

According to the described hypothetical model, Gd-IgA1 and the immune complexes containing it are involved in the pathogenesis of IgAV, particularly nephritis. Gd-IgA1 likely contributes to systemic inflammation by forming circulating immune complexes, which then deposit in small blood vessels. These complexes activate the alternative complement pathway and, by binding to CD89 on neutrophil granulocytes, release enzymes and ROS that ultimately damage the endothelium [10,44,54,55]. The results of our study did not show a significant difference in serum Gd-IgA1 levels between patients and controls, nor among IgAV patients themselves regarding nephritis. However, Gd-IgA1 appears to be involved in arthritis as a clinical manifestation of the disease since it has been highlighted as a predictor. The study unveiled a significant difference in urine concentrations of Gd-IgA1 between patients and controls, which nevertheless suggests potential involvement in IgAV. Although there was no difference between patients regarding nephritis, the observed positive correlations of Gd-IgA1, particularly with the laboratory parameters of renal function, are worth emphasizing, probably due to its indirect association with nephritis in IgAV.

The results of this study also suggest a potential role of HMGB1 in the pathogenesis of IgAV, as serum and urine concentrations of HMGB1 were significantly different between patients and controls. A limited number of studies on IgAV have shown that concentrations of HMGB1 were notably higher in the serum of children with IgAV and that it was abundantly expressed in the cytoplasm of human dermal microvascular endothelial cells. The results further support the association of HMGB1 with the pathogenesis of IgAV, suggesting its involvement in the second hypothesis model of the disease, where AECA and the activation of endothelial cells play key pathomechanistic roles in endothelial damage. HMGB1 is most likely released from activated neutrophil granulocytes, thereby enhancing inflammation by activating the NF-κB, a central signaling pathway involved in proinflammatory cytokine gene transcription, and also increasing the expression of endothelial adhesion molecules ICAM-1, VCAM-1 and selectins [30,56]. The role of HMGB1 in the inflammatory response in IgAV is evident and furthermore supported by significant positive correlations with CRP, ferritin and IgG, all of which are considered important inflammatory parameters in autoimmune rheumatic diseases. In addition to the observed difference between patients with IgAV and the controls, serum concentrations of HMGB1 significantly differed at the onset of the disease and after a six-month follow-up interval. This may imply that HMGB1 has a certain impact on the development of IgAV itself, given its higher concentrations in the acute phase when the disease appears, followed by an expected recovery from the disease. Therefore, HMGB1 could be used to monitor patients with IgAV. HMGB1 certainly plays an important role in IgAV, as logistic regression identifies it as significant in predicting IgAV and arthritis as well. Concentrations of HMGB1 in urine were higher in IgAV patients with nephritis compared to patients without nephritis in the follow-up period and showed positive correlations with the findings of erythrocyturia in urine tests, uACR and with the outcome of nephritis. Since a certain number of patients will develop nephritis within the clinical course of IgAV, measuring the concentration of HMGB1 in the urine may be a useful tool in evaluating IgAV patients with nephritis and in making therapeutic approaches.

Higher concentrations of sRAGE were found in the serum and urine of patients with IgAV, even after performing logistic regression with model adjustments, which therefore provides strong support to speculate that RAGE plays a proinflammatory role in the development of IgAV. Immune complexes can activate the expression of RAGE on human endothelial cells and participate in the response of the HMGB1-RAGE axis in promoting IgAV and inducing the production of TNF-α. Moreover, by binding to the β2-integrin Mac-1 (CD-11b/CD18) on endothelial cells, sRAGE activates the transcription factor NF-κB through the MAPK signaling pathway and enhances the inflammatory response through the promotion of chemotaxis and leukocytes recruitment [57,58]. Currently, there are no other studies and evidence about the involvement of RAGE in IgAV; therefore, its mechanism requires further investigation. Among other results, it is worth emphasizing the significant positive correlation of RAGE with musculoskeletal manifestations of IgAV, including arthritis and arthralgias, respectively. It is possible that RAGE contributes to the pathogenesis of IgAV, as logistic regression highlights serum RAGE as a predictor of arthritis. There were no differences in serum and urine concentrations of RAGE between patients with and without nephritis to report on, but significant positive correlations with uACR and 24 h proteinuria values suggest a possible contribution to nephritis in IgAV. Similar to serum HMGB1, the serum concentrations of RAGE differed between IgAV patients at baseline and after the follow-up interval. This may indicate residual low inflammatory activity or tissue damage despite clinical remission, suggesting that RAGE could have potential in disease activity evaluation and even as a therapeutic target.

It is known that PCDH1 contributes to the maintenance of intercellular junctions and provides a certain elasticity to epithelial tissues. The binding of PCDH1 to the signal molecule SMAD3 seems to suppress the TGF-β signaling pathway [38]. Activation of the TGF-β-SMAD3 signaling pathway is required for the differentiation of regulatory T lymphocytes. After binding with TGF-β, SMAD3 enters into the nucleus where it interacts with transcription factors that suppress the production of IL-2 and IFN-γ and stimulate the transcription of Foxp3 [59,60,61]. A reduced level of regulatory T lymphocytes is associated with the development of IgAV in children, so it is possible that this is precisely the pathogenic pathway by which PCDH1 is involved in the pathogenesis of IgAV [62,63,64,65]. Results support a role of PCDH1 in the development of IgAV since serum and urine concentrations were significantly higher in patients than in the control group. Logistic regression highlights serum PCDH1 as significant in the prediction of arthritis, and positive correlations were also found; therefore, it is possible that it is involved in the pathogenesis of musculoskeletal manifestations of the disease. Similar to the above-described molecules, in the case of urinary PCDH1, a continuous positive correlation was also found with uACR, a laboratory parameter used in daily clinical work to assess and monitor patients with nephritis.

This study has several limitations. The sample is relatively small, especially the number of patients with nephritis, so the associations and significances that might exist for observation are limited. Additionally, due to technical restrictions, it was not possible to measure all laboratory parameters, which could potentially have an impact on some of the results. Moreover, every practical laboratory test has its accuracy and limitations, including immunoenzymatic tests, so the results of the study may be affected by the complex technical and methodological steps of the investigation itself. The effect of anti-inflammatory drugs should also be taken into consideration, since in a certain number of patients, the medications were administered before collecting serum and urine samples, and thus could have interfered with the results of the study.

## 4. Materials and Methods

### 4.1. Participants and Study Design

The present study was designed as a prospective trial and conducted from January 2020 to October 2023 in three Croatian pediatric rheumatology centers.

A total of 156 subjects under 18 years of age were enrolled, including 86 diagnosed with IgAV and 70 from the control group. All IgAV patients fulfilled the European League Against Rheumatism (EULAR)/Pediatric Rheumatology International Trials Organization (PRINTO)/Pediatric Rheumatology European Society (PRES) classification criteria [2]. Controls were screened for the presence of systemic autoimmune diseases or any other symptoms suggestive of an inflammatory condition.

The study complied with the principles of the Declaration of Helsinki and was approved by the Ethics Committee of University of Zagreb (date: 18 September 2019; Protocol Number-Class: 641-01/19-02/01) for clinical studies involving human subjects. Written informed consent was provided from the parents or legal guardians of minors in all cases.

### 4.2. Clinical Assessment

A detailed medical history, along with systematic and rheumatologic examinations, were performed on all IgAV patients at the onset of the disease and after a six-month interval. The following clinical characteristics were observed in patients: prevalence, distribution and severity of skin changes; prevalence of arthritis and/or arthralgias; and involvement of the gastrointestinal, renal and urogenital systems. Blood pressure was measured, and data on administered medications were taken for each child.

### 4.3. Disease Activity Assessment

The Pediatric Vasculitis Activity Score (PVAS) was used to record clinical features of IgAV during the active phase of the disease and was performed by a rheumatologist. PVAS consists of 64 clinical variables divided into nine organ systems (general symptoms, cutaneous, mucous membranes/eyes, ear/nose/throat, chest, cardiovascular, abdominal, renal and central and/or peripheral nervous system) and follows a format similar to that of the Birmingham Vasculitis Activity Score (BVAS) [66,67]. Each item, if present, is evaluated with a certain score, and the sum of the scores of the nine organ systems determines the total score, quantifying the level of disease activity of the patient at the time of the assessment with a total score ranging from 0 to 63. The outcome of nephritis was assessed based on the modified Counahan classification in patients who developed nephritis after a six-month follow-up interval [68].

### 4.4. Laboratory Analysis

After obtaining clinically relevant data from medical records, blood and urine samples for routine laboratory tests were collected from both patients and controls.

The laboratory findings measured and compared between patients at the onset of the disease and controls included the following: inflammatory markers [erythrocyte sedimentation rate (ESR) and C-reactive protein (CRP)], complete blood count (erythrocytes, hemoglobin, leukocytes, platelets), biochemical blood tests (creatinine, urea, total protein levels, serum albumin), coagulation factor tests [fibrinogen, D-dimer test, prothrombin time (PT), activated partial thromboplastin time (aPTT)], immunoassays (immunoglobulin classes: IgA, IgG, IgM), spot urine sample tests and the urine albumine to creatinine ratio (uACR).

Additionally, in patients with IgAV several other tests were performed, including ferritin, complement components C_3_ and C_4_, total complement activity (CH50), fecal calprotectin levels, fecal occult blood tests (FOBT) and analysis of a 24 h urine protein test. All the listed laboratory tests were repeated after a six-month interval only in IgAV patients at regular follow-up rheumatologic visits.

### 4.5. Analysis of Gd-IgA1, HMGB1, RAGE and PCDH1

Amounts of 6 mL of cubital venous blood and 6 mL of urine were taken for the analysis of Gd-IgA1, HMGB1, RAGE and PCDH1 on the same day when routine laboratory tests were performed in patients with IgAV at the onset of the disease and after a six-month interval visit, and once in the control group as well. The sampled blood and urine were centrifuged at 2000–3000 rpm for 10–15 min (*Hettich Torofix 32; Andreas Hettich GmbH & Co. KG, Tuttlingen, Germany*) at room temperature, after which aspired aliquotes were separated into cryotubes (*Kartell, Noviglio, Italy*) and stored at −80 °C (*Eppendorf U410; s.br.: F410IR113753; Eppendorf, Hamburg, Germany*) until use. Serum and urine concentrations of Gd-IgA1, HMGB1, RAGE and PCDH1 were determined by enzyme-linked immunosorbent assay (ELISA), using commercially-available diagnostic kits [*Human Gd-IgA1 ELISA MyBioSource (San Diego, SAD, CA, USA) kit Cat.NO MBS1607395; SEA399Hu 96 tests ELISA High Mobility Group Protein 1 (HMG1) Cloud Clone Corp. (Houston, SAD, TX, USA)*; *Human RAGE Elisa kit Biorbyt (Cambridge, UK) Cat.NO orb864692*; *Human PCDH1 ELISA kit Biorbyt (Cambridge, UK)*], following the manufacturer’s instructions. The results of the enzyme immunoassay were read using a SPECTROstar^®^ Nano (*BMG Labtech GmbH, Ortenberg, Germany*, serial number 601-1808) microplate spectrophotometer.

### 4.6. Statistical Analysis

Statistica for Windows version 12.0, GraphPad Prism version 9.1.2 and R for Windows version 4.2.3. were used for statistical analysis [69,70]. Continuous data are presented as the mean±standard deviation (SD) or median [interquartile range (IQR)] based on the variable distribution normality, whereas categorical data are expressed as numbers (N) with percentages (%). Normality of distribution for continuous variables was assessed using the Shapiro–Wilk test. For differences between the groups, an independent samples t-test was used for continuous variables with normal distribution, whereas the Mann–Whitney U test with Bonferroni correction for multiple comparisons was employed for the analysis of continuous variables with non-normal distribution. The Wilcoxon signed-rank test for dependent variables was performed to analyze differences between patients with IgAV at baseline and after the follow-up interval. The chi-squared (χ^2^) test or Fisher exact test was used to determine differences between groups in terms of categorical variables. To investigate the correlation between Gd-IgA1, HMGB1, RAGE and PCDH1 and clinical and laboratory parameters and PVAS, we used Kendall’s tau correlation coefficient. Logistic regression analysis was used to find which of the evaluated molecules could be a predictor of IgAV or clinical features of the disease. ROC analysis with calculated AUC was constructed to evaluate the significant optimal cut-off value with highest predictive accuracy for Gd-IgA1, HMGB1, RAGE and PCDH1 in discriminating between patients and controls [71]. All *p*-values were two-tailed, and the level of statistical significance was set at *p* < 0.05.

## 5. Conclusions

In conclusion, the presented results suggest that Gd-IgA1, HMGB1, RAGE and PCDH1 interplay in the complex pathogenesis of IgAV and influence certain clinical features of the disease, likely based on their roles in inflammatory and reparative processes in human tissues. Moreover, some of the assessed molecules may be involved in the development of nephritis, the most prominent chronic manifestation of the disease itself. While the overall results of our study should be interpreted cautiously, urinary HMGB1 particularly highlighted its potential. Since identifying a useful surrogate biomarker for nephritis in IgAV remains challenging, further larger scale studies are desirable and necessary to fully address this issue.

## Figures and Tables

**Figure 1 ijms-25-04383-f001:**
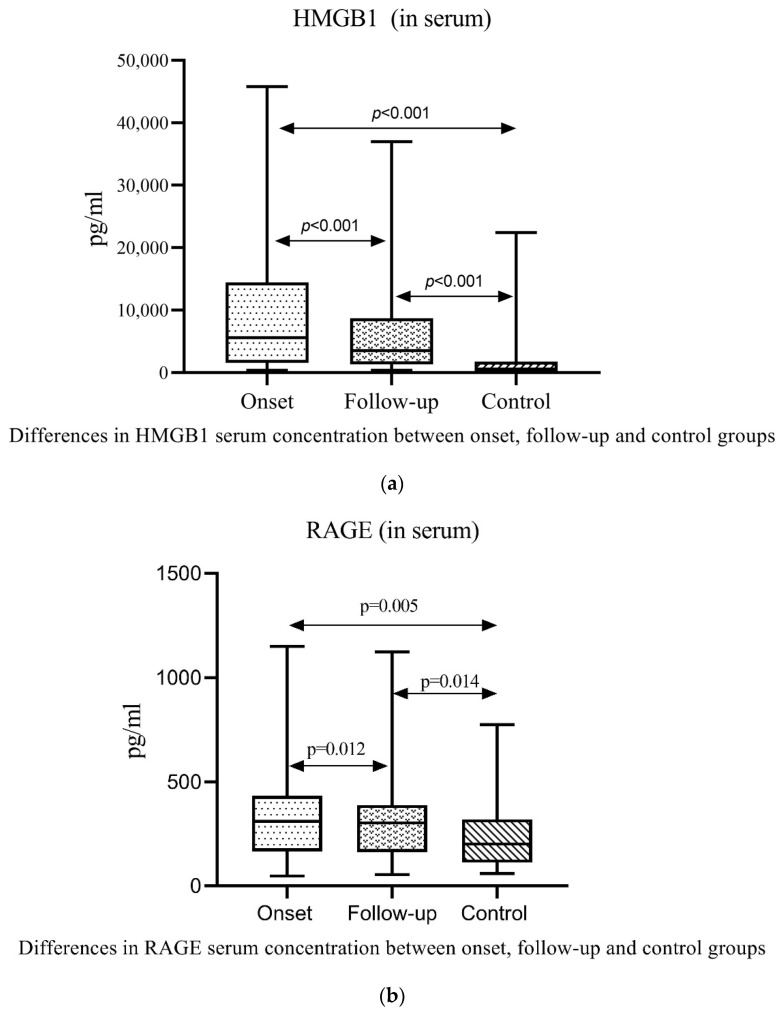
Box plot display of serum concentrations of HMGB1 and RAGE in patients with IgAV and the control group detected by ELISA. Data were presented as min-average-max. The data between groups was analyzed using the Mann–Whitney U test (with Bonferroni correction) for independent samples, while matched samples were analyzed using the Wilcoxon signed-rank test. (**a**) IgAV onset vs. control group, *p* < 0.001; IgAV follow-up vs. control group, *p* < 0.001; IgAV onset vs. IgAV follow-up group, *p* < 0.001; (**b**) IgAV onset vs. control group, *p* = 0.005; IgAV follow-up vs. control group, *p* = 0.014; IgAV onset vs. IgAV follow-up group, *p* = 0.012.

**Table 1 ijms-25-04383-t001:** Baseline characteristics of patients with IgAV.

	N = 86	% of the Cohort
**Demographic data**		
mean age (years)	6.4 (4.5–7.8)	
female	43	57%
male	37	43%
**Clinical features**		
skin changes	86	100%
joint involvement	76	88.4%
gastrointestinal involvement	39	45.3%
nephritis	26	30.2%
scrotal involvement	6	16.2% ^#^
**Treatment**		
NSAIDs	75	87.2%
Glucocorticoids	36	41.8%
ACE inhibitors	15	17.4%
immunosuppressants	11	12.8%
**PVAS**	4 (2–6)	
**outcome of nephritis**		
A	18	69.3% ^##^
B	8	30.7% ^##^
C	0	0%
D	0	0%

Data are presented as a whole number and percentages (%); ^#^ applicable only for boys; ^##^ applicable only for patients with nephritis; NSAID: non-steroidal anti-inflammatory drugs; ACE: angiotensin converting enzyme; PVAS: Pediatric Vasculitis Activity Score; A: normal on physical examination, with normal urine or microhematuria and normal renal function; B: normal on physical examination, with proteinuria < 1 g per day or <40 mg/h per m^2^, urine albumine to creatinine ratio < 200 mg/mmol; C: proteinuria > 1 g per day or >40 mg/h/m^2^ and/or hypertension, glomerular filtration rate > 60 mL/min/1.73 m^2^; D: renal insufficiency (estimated glomerular filtration rate < 60 mL/min/1.73 m^2^ or end stage renal disease (estimated glomerular filtration rate < 15 mL/min/1.73 m^2^) or death.

**Table 2 ijms-25-04383-t002:** Laboratory findings in the group of patients with IgAV and the control group.

Parameter	IgAV Group		Control Group	*p* *	*p* **	*p* ***
	Onset	Follow Up				
ESR (mm/h)	18 (8–30)	9 (6–14)	7 (4.75–11)	**<0.001**	**0.012**	**<0.001**
CRP (mg/L)	6.8 (2.1–18.3)	1 (0.6–1.7)	1 (1–1)	**<0.001**	0.406	**<0.001**
leukocytes (10^9^/L)	10.1 (8.2–12.8)	7.24 (6.1–8.03)	6.15 (5.0–7.0)	**<0.001**	**<0.001**	**<0.001**
neutrophils	5.5 (4.2–8.1)	3.15 (2.41–4.04)	2.7 (2.1–3.7)	**<0.001**	0.889	**<0.001**
erythrocytes (10^12^/L)	4.62 ± 0.42	4.75 ± 0.35	4.75 ± 0.38	0.063	0.078	**0.04**
hemoglobin (g/L)	124.3 ± 11.3	127.2 ± 8.9	130.8 ± 11.1	**<0.001**	**0.015**	0.068
platelets (10^9^/L)	355.3 ± 82.1	334.1 ± 68.5	291.8 ± 71.6	**<0.001**	**<0.001**	0.072
creatinine (µmol/L)	35 (28–40.25)	36 (31.5–42)	44 (36.25–50.75)	**<0.001**	**<0.001**	0.197
urea (mmol/L)	4.5 ± 1.15	4.12 ± 1.01	4.4 ± 1.08	0.647	0.462	**0.005**
fibrinogen (g/L)	3.5 (2.8–4.2)	2.7 (2.4–3.1)	2.4 (2.2–2.8)	**<0.001**	0.118	**<0.001**
D-dimer (µg/L)	2.9 (1.1–7.3)	0.27 (0.19–0.36)	0.2 (0.19–0.33)	**<0.001**	0.865	**<0.001**
PT	1.01 ± 0.15	1.03 ± 0.16	0.99 ± 0.13	0.653	0.153	0.624
aPTT (s)	25.4 ± 3.8	27.4 ± 2.8	25.65 ± 2.09	0.703	0.425	**<0.001**
ferritin (ng/mL)	58.45 (34.6–86.1)	24.85 (16.2–41.7)	-	-	-	**<0.001**
uACR (mg/mmol)	5.1 (1.3–14)	3.7 (1–12)	0.85 (0.6–1.3)	**<0.001**	**<0.001**	0.154
24-h proteinuria (g/dU)	0.07 (0.05–0.15)	0.07 (0.05–0.14)	-	-	-	0.625
E/mm^3^ (urine spot test) (%)	22 (25.6%)	13 (15.5%)	-	-	-	0.103
proteinuria (urine spot test) (%)	14 (16.3%)	8 (9.5%)	-	-	-	0.189
eGFR (mL/min/1.73 m^2^)	139.2 ± 29.7	148.2 ± 29.4	-	-	-	0.111
total proteins (g/L)	68.5 (65–72)	72 (67–74)	72 (69.5–74)	**<0.001**	0.289	**<0.001**
serum albumin (g/L)	39.44 ± 5.13	43.95 ± 4.84	45.16 ± 3.05	**<0.001**	0.298	**<0.001**
IgA (g/L)	1.7 (1.25–2.42)	1.49 (1–1.9)	1.24 (0.98–1.56)	**<0.001**	0.472	**<0.001**
IgG (g/L)	10.92 ± 2.92	10.1 ± 2.16	10.68 ± 2.09	0.612	0.392	**<0.001**
IgM (g/L)	0.94 (0.7–1.2)	1.03 (0.76–1.26)	1.1 (0.8–1.3)	0.164	0.968	**<0.001**
C_3_ (g/L)	1.32 ± 0.24	1.22 ± 0.21	-	-	-	
C_4_ (g/L)	0.27 (0.21–0.33)	0.23 (0.17–0.26)	-	-	-	**<0.001**
CH50 (%)	91 (79–109)	81 (76–97)	-	-	-	
positive FOBT (%)	24 (27.9%)	1 (1.2%)	-	-	-	**<0.001**
fecal calprotectin (µg/g)	37 (20–83)	28 (20–34)				

* IgAV (onset) vs. controls; ** IgAV (follow-up) vs. controls; *** IgAV (onset) vs. IgAV (follow-up). Data are presented as the whole number (percentage), mean ± standard deviation (SD) or median (IQR); statistical significance was set to *p* < 0.05 and bolded; statistical tests used in analysis were the Mann–Whitney U test, *t*-test for independent samples, Wilcoxon signed-rank test for dependent data, chi-square test and Fisher exact test. Legend: IgAV—IgA vasculitis; ESR—erythrocyte sedimentation rate; CRP—C-reactive protein; PT—prothrombin time; aPTT—activated partial thromboplastin time; uACR—urine albumine to creatinine ratio; IgA—immunoglobulin A; IgG—immunoglobulin G; IgM—immunoglobulin M; C_3_—complement component C_3_; C_4_—complement component C_4_; CH50—total complement activity; FOBT—fecal occult blood test.

**Table 3 ijms-25-04383-t003:** Serum and urine concentrations of Gd-IgA1, HMGB1, RAGE and PCDH1 in patients with IgAV and the control group.

Parameter		IgAV Group		Control Group (N = 70)	*p* *	*p* **	*p* ***
		Onset (N = 86)	Follow-Up (N = 83)				
**Gd-IgA1 (µg/mL)**	serum	52.4 (25.3–96.1)	49.9 (22.1–87.8)	54.1 (19.4–90.3)	0.299	0.668	0.231
	urine	10.8 (6.3–21.2)	11.7 (5.9–18.9)	5.4 (3.2–9.3)	**<0.001**	**<0.001**	0.944
**HMGB1 (pg/mL)**	serum	5573 (2274–13,829)	3477 (1308–6445)	1088 (574.3–2942)	**<0.001**	**<0.001**	**<0.001**
	urine	178.4 (96.7–596.9)	173.1 (94.2–380.2)	57.1 (36.9–168)	**<0.001**	**<0.001**	0.755
**RAGE (pg/mL)**	serum	309.9 (166.6–432.1)	302.4 (163.2–388)	201.3 (112.7–319.6)	**0.005**	**0.014**	**0.012**
	urine	12.2 (7.3–21.2)	10.1 (5.9–18.7)	5.9 (4.1–10.4)	**<0.001**	**<0.001**	0.077
**PCDH1 (ng/mL)**	serum	44.5 (28.2–61.5)	46.9 (31.8–59.6)	18.8 (1.2–45.4)	**<0.001**	**<0.001**	0.746
	urine	174.6 (72.8–327.5)	226.4 (102.1–414.9)	71.1 (2.1–138.1)	**<0.001**	**<0.001**	0.109

* IgAV (onset) vs. controls; ** IgAV (follow-up) vs. controls; *** IgAV (onset) vs. IgAV (follow-up). Data are presented as median (IQR); statistical significance was set to *p* < 0.05 and bolded; Mann–Whitney U test was used for independent samples; Wilcoxon signed-rank test was used for dependent data (IgAV onset vs. follow-up). Legend: IgAV—IgA vasculitis; Gd-IgA1—galactose-deficient immunoglobulin A1; HMGB1—high-mobility group protein box 1; RAGE—receptor for advanced glycation end products; PCDH1—protocadherin-1.

**Table 4 ijms-25-04383-t004:** Serum and urine concentrations of Gd-IgA1, HMGB1, RAGE and PCDH1 in patients with IgAV without nephritis and with nephritis.

Parameter		IgAV Group		IgAVN Group		*p* *	*p* **
		Onset (N = 60)	Follow-Up (N = 58)	Onset (N = 26)	Follow-Up (N = 25)		
**Gd-IgA1 (µg/mL)**	serum	55.7 (23.5–96.9)	51.09 (21.52–88.75)	50 (31.2–91.6)	40.24 (27.93–90.1)	0.783	0.863
	urine	10.8 (6.3–18.9)	12.04 (6.86–19.36)	10.8 (6.4–25.7)	11.17 (4.26–19.98)	0.657	0.618
**HMGB1 (pg/mL)**	serum	5477.5 (2266.6–14,531.6)	3245 (1367–5552)	6303.9 (2349–11,042.9)	3519 (1241–9295)	0.966	0.832
	urine	143.1 (88.9–568.3)	133.2 (85.9–318.6)	340.9 (134.7–644.1)	270.9 (146.7–542.7)	0.083	**0.049**
**RAGE (pg/mL)**	serum	294.1 (145.3–434.2)	275.9 (140.8–369)	314.1 (251.7–425.2)	331.3 (236.4–416.6)	0.492	0.073
	urine	12.2 (8.2–17.6)	8.6 (5.9–19.1)	13.8 (6.9–29.1)	11.99 (6.04–18.52)	0.540	0.697
**PCDH1 (ng/mL)**	serum	46.5 (32.4–62.4)	49.74 (38.35–59.07)	41.0 (2.6–59)	41.27 (10.8–65.1)	0.309	0.206
	urine	174.4 (86.3–302.7)	209.1 (103–409.5)	186.4 (6.2–364.1)	257.1 (77.6–459.2)	0.733	0.699

* IgAV (onset) vs. IgAVN (onset); ** IgAV (follow-up) vs. IgAVN (follow-up). Data are presented as median (IQR); statistical significance was set to *p* < 0.05 and bolded; Mann–Whitney U test was used for independent samples. Legend: IgAV—IgA vasculitis; Gd-IgA1—galactose-deficient immunoglobulin A1; HMGB1—high-mobility group protein box 1; RAGE—receptor for advanced glycation end products; PCDH1—protocadherin-1.

**Table 5 ijms-25-04383-t005:** AUC and optimal cut-off value of serum and urine concentrations of Gd-IgA1, HMGB1, RAGE and PCDH1 in the identification of patients with IgAV from the control group.

Parameter		AUC	95% CI	*p* *	Youden Index	Cut-Off Value	Sensitivity	Specificity
**Gd-IgA1 (µg/mL)**	serum	0.549	0.467–0.628	0.3041	0.1575	>31.716	68.60	47.14
	urine	0.736	0.659–0.803	**<0.0001**	0.4130	>6.383	75.58	65.71
**HMGB1 (pg/mL)**	serum	0.784	0.711–0.846	**<0.0001**	0.5030	>3926.95	67.44	82.86
	urine	0.782	0.709–0.844	**<0.0001**	0.4944	>81.481	83.72	65.71
**RAGE (pg/mL)**	serum	0.631	0.550–0.707	**0.0037**	0.2508	>329.02	46.51	78.57
	urine	0.720	0.642–0.789	**<0.0001**	0.4146	>9.254	68.60	72.86
**PCDH1 (ng/mL)**	serum	0.705	0.625–0.776	**<0.0001**	0.3496	>29.03	75.58	59.38
	urine	0.728	0.650–0.798	**<0.0001**	0.3590	>158.18	54.65	81.25

Legend: Gd-IgA1—galactose-deficient immunoglobulin A1; HMGB1—high-mobility group protein box 1; RAGE—receptor for advanced glycation end products; PCDH1—protocadherin-1; AUC—area under the curve; CI—confidence interval; statistical significance was set to * *p* < 0.05 and bolded.

**Table 6 ijms-25-04383-t006:** Logistic regression for the prediction of IgAV.

Parameter		SE	Z-Value	OR	95% CI	*p* *
**Gd-IgA1 (µg/mL)**	serum	0.107	0.555	1.001	0.997–1.006	0.579
	urine	0.341	1.159	1.029	0.981–1.087	0.246
**HMGB1 (pg/mL)**	serum	0.036	3.923	1.000	1.0001–1.0003	**<0.001**
	urine	0.083	−0.122	0.999	0.999–1.001	0.903
**RAGE (pg/mL)**	serum	0.074	2.712	1.004	1.001–1.007	**0.007**
	urine	0.224	1.968	1.033	1.002–1.070	**0.049**
**PCDH1 (ng/mL)**	serum	0.487	1.849	1.018	0.999–1.038	0.064
	urine	0.098	0.495	1.001	0.997–1.005	0.621

Legend: Gd-IgA1—galactose-deficient immunoglobulin A1; HMGB1—high-mobility group protein box 1; RAGE—receptor for advanced glycation end products; PCDH1—protocadherin-1; SE—standard error; CI—confidence interval. Statistical significance was set to * *p* < 0.05 and bolded.

## Data Availability

All data are available in the manuscript. The datasets generated during and/or analyzed during the current study are available from the corresponding author on reasonable request.

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
