# Peer review of "Insight into the Interplay of Gd-IgA1, HMGB1, RAGE and PCDH1 in IgA Vasculitis (IgAV)"

_ijms, 2024, doi:10.3390/ijms25084383_

Round 1

Reviewer 1 Report

Comments and Suggestions for Authors

The article "An Examination of the Interplay between Gd-IgA1, HMGB1, RAGE, and PCDH1 in IgA Vasculitis (IgAV)" examines the relationship between several biomarkers and IgA vasculitis. While the study presents commendable research efforts, significant revisions are necessary to enhance clarity, reliability, and overall quality.

The abstract lacks critical aspects such as a clear description of study objectives, methods, noteworthy findings, and implications. To address this matter, it is recommended to arrange the abstract in a manner that adheres to a prescribed structure encompassing the background, methodology, findings, and conclusions. In order to highlight its significance within the existing literature, it is essential to explain how the research on IgA vasculitis is unique and how it has an effect.

The introduction lacks adequate citations and references, impeding the development of the research framework and the validation of the study's importance. Strengthen the background information by doing a thorough analysis of relevant publications. To develop a clear and focused purpose, it is crucial to accurately characterize the current research gap and explain how the current study aims to address it.

Improvement is needed in language and citation within the introduction section to strengthen clarity and rationale for the study. Engaging a professional English editor and academic writer may be beneficial.

Inclusion of central tendency measurements, sample sizes, and statistical significance in Table 4 and Figure 1 would enhance reader comprehension and resilience, providing valuable context for interpreting the findings.

The caption of Figure 1 lacks essential information, such as specific statistical tests and adjustments for multiple comparisons, undermining the reliability of the reported results. Clear and complete captions are essential for readers to assess the validity of the findings accurately.

Insufficient results presentation in Table 4 and Figure 1 compromises study reliability and reproducibility. It is recommended to include metrics of central tendency, sample sizes, and statistical significance to enhance transparency and facilitate comprehension of data.

There has not been enough critical analysis or in-depth review of the study's results. To improve the analysis, it is advisable to provide a more comprehensive examination of the results, exploring potential mechanisms that underlie the observed associations and their significance in understanding the pathophysiology of IgA vasculitis. Furthermore, the conclusions must offer meaningful advice for future research and therapeutic use.

The statistical procedures utilized in the analysis offered in the account are insufficient for replication and confirmation by other researchers. Please explain each statistical test used, including the implicit assumptions and the rationale for their choice. It is recommended to get help from a statistical expert to evaluate the appropriateness and robustness of the statistical methods utilized in the study.

The establishment of external validity is crucial to generalize the results of a study to a wider population beyond the specific sample being examined. Please provide a detailed explanation of the demographic and clinical characteristics of the research group and examine their importance in terms of broader relevance. It is recommended to conduct additional analysis or subgroup analysis to examine potential variations in the study findings across different groups or circumstances.

The study's internal validity and susceptibility to bias are compromised due to the lack of explicit criteria for selecting the control group. The establishment of comparability with patients diagnosed with IgA vasculitis relies heavily on the criteria utilized for the selection of control individuals. To address potential confounding factors, it is recommended to match control participants with individuals diagnosed with IgA vasculitis, while considering relevant demographic and clinical attributes.

When writing scientifically, it is recommended to alter instances where the pronoun "we" is employed in the text to maintain impartiality and professionalism. Conversely, choose for more appropriate wording such as "the study unveiled" or "the findings suggest." To enhance the manuscript's integrity, it is necessary to address these specific cases. The subsequent alterations are advised for the specified lines:

- Line 336: "...and therefore provides strong support..."

The statement made in line 338 suggests that it may contribute to the progression of IgA vasculitis and initiate...

The activation of sRAGE takes place on endothelial cells.

According to line 341 of the text, it enhances the inflammatory response by facilitating...

The unambiguous manifestation of HMGB1's participation in the inflammatory response in IgAV is evident.

According to line 319 of the book, there are significant positive correlations that offer more evidence.

Furthermore, it is worth noting that there is a noticeable difference in blood concentrations between individuals diagnosed with IgAV and the control group.

Line 322 states that the onset of the sickness occurred after a period of six months of observation. These findings indicate that HMGB1...

In line 323, it is stated that the high levels of IgAV have a special effect on its growth.

In line 324, the user mentions the early stage of disease manifestation, which is followed by an expected recovery.

Comments on the Quality of English Language

Extensive English Revision is required                                                                                                                                                       To maintain objectivity and professionalism in scientific writing, it is recommended to revise instances where the term "we" is used in the manuscript. Instead, opt for more appropriate language such as "the study found" or "the results suggest." Addressing these instances will enhance the integrity of the manuscript. Here are the suggested revisions for the identified lines:

- Line 336: "...and therefore provides strong support..."

- Line 338: "...promoting IgA vasculitis and inducing..."

- Line 340: "...on endothelial cells, sRAGE activates..."

- Line 341: "...enhances the inflammatory response through promotion..."

- Line 318: "...role of HMGB1 in the inflammatory response in IgAV is evident..."

- Line 319: "...furthermore supported by significant positive correlations..."

- Line 321: "...In addition to the observed difference between IgAV patients and controls, serum concentrations..."

- Line 322: "...the onset of the disease and after a six-month follow-up interval. This may imply that HMGB1..."

- Line 323: "...has a certain impact on the development of IgAV itself, given its higher concentrations..."

- Line 324: "...the acute phase when disease appears, followed by an expected recovery..."

Addressing these recommendations comprehensively will significantly improve the manuscript's quality and impact, making it more suitable for publication and contributing meaningfully to the field of IgA vasculitis research.

Author Response

1. Summary

We would like to thank Reviewer 1 for taking the time to review this manuscript and for suggestions how to improve it. Please find the detailed responses below and the corresponding revisions/corrections highlighted/in track changes in the re-submitted files.

2. Questions for General Evaluation

Reviewer’s Evaluation

Response and Revisions

Does the introduction provide sufficient background and include all relevant references?

Must be improved

We have modified introduction as explained in Response 2.

Are all the cited references relevant to the research?

Must be improved

Is the research design appropriate?

Can be improved

Are the methods adequately described?

Can be improved

Are the results clearly presented?

Must be improved

Are the conclusions supported by the results?

Must be improved

3. Point-by-point response to Comments and Suggestions for Authors

Comments 1:

The abstract lacks critical aspects such as a clear description of study objectives, methods, noteworthy findings, and implications. To address this matter, it is recommended to arrange the abstract in a manner that adheres to a prescribed structure encompassing the background, methodology, findings, and conclusions. In order to highlight its significance within the existing literature, it is essential to explain how the research on IgA vasculitis is unique and how it has an effect.

Response 1: Thank you for Your comment, however we think that abstract of our manuscript is correctly written. When preparing the abstract for the paper, we adhered the abstract formatting instructions published on the IJMS website, which stated that the abstract should follow the style of structured abstracts, but without usual headings included. Since the number of words in the abstract is limited, we are not able to further increase it. The importance of research of IgAV and all the unknowns related to the pathogenesis of the disease itself, was emphasized in the summary.

Comments 2: The introduction lacks adequate citations and references, impeding the development of the research framework and the validation of the study's importance. Strengthen the background information by doing a thorough analysis of relevant publications. To develop a clear and focused purpose, it is crucial to accurately characterize the current research gap and explain how the current study aims to address it. Improvement is needed in language and citation within the introduction section to strengthen clarity and rationale for the study. Engaging a professional English editor and academic writer may be beneficial.

Response 2: The aim of our study was to investigate potential markers of IgAV, to investigate the pathogenesis of the disease with a special emphasis on nephritis, since identifying markers that may predict nephritis is still the most challenging area of research in IgAV. Therefore, we don’t agree with the comment that the introduction of our manuscript lacks basic overview since we indeed tried to cover the topic in a reviewed, clear, and meaningful way. We agree with the comments regarding English language, so we carefully went through manuscript in spelling and written form for overall intelectual content.

Comments 3: Inclusion of central tendency measurements, sample sizes, and statistical significance in Table 4 and Figure 1 would enhance reader comprehension and resilience, providing valuable context for interpreting the findings. The caption of Figure 1 lacks essential information, such as specific statistical tests and adjustments for multiple comparisons, undermining the reliability of the reported results. Clear and complete captions are essential for readers to assess the validity of the findings accurately. Insufficient results presentation in Table 4 and Figure 1 compromises study reliability and reproducibility. It is recommended to include metrics of central tendency, sample sizes, and statistical significance to enhance transparency and facilitate comprehension of data.

Response 3: We understand your comment, however we did not consider necessary to include central tendency measurements and statistical significance in Figure 1 since the results in this regard are already shown in detail in Table 3. As can be seen, results in Table 3. are shown as median and interquartile range with significant p values marked with bold font. Figure 1 is an additional graphic presentation of the above results from Table 3.

Comments 4: There has not been enough critical analysis or in-depth review of the study's results. To improve the analysis, it is advisable to provide a more comprehensive examination of the results, exploring potential mechanisms that underlie the observed associations and their significance in understanding the pathophysiology of IgA vasculitis. Furthermore, the conclusions must offer meaningful advice for future research and therapeutic use.

Response 4: Thank you for suggestion. However, we believe that we addressed all observed associations in our conclusion properly. Regarding results, we tried to show everything relevant to this study, so for that reason we correlated each molecule with clinical features and laboratory findings, provide univariate logistic regression and perform ROC analysis. We believe that the results presented in our manuscript are more than sufficiently.

Comments 5: The statistical procedures utilized in the analysis offered in the account are insufficient for replication and confirmation by other researchers. Please explain each statistical test used, including the implicit assumptions and the rationale for their choice. It is recommended to get help from a statistical expert to evaluate the appropriateness and robustness of the statistical methods utilized in the study.

Response 5: We have accordingly extended the results by adding ROC curve analysis presented in Table 5.

Comments 6: The establishment of external validity is crucial to generalize the results of a study to a wider population beyond the specific sample being examined. Please provide a detailed explanation of the demographic and clinical characteristics of the research group and examine their importance in terms of broader relevance. It is recommended to conduct additional analysis or subgroup analysis to examine potential variations in the study findings across different groups or circumstances.

Response 6: We believe that we have included an acceptable number of subgroups. Principally, we analyzed and compared patients with IgAV and the control group, then we compared patients with IgAV at the onset of the disease and after six months interval, and finally we compared patients with IgAV regarding on they had nephritis or not.

Comments 7: The study's internal validity and susceptibility to bias are compromised due to the lack of explicit criteria for selecting the control group. The establishment of comparability with patients diagnosed with IgA vasculitis relies heavily on the criteria utilized for the selection of control individuals. To address potential confounding factors, it is recommended to match control participants with individuals diagnosed with IgA vasculitis, while considering relevant demographic and clinical attributes.

Response 7: We matched the patients with IgAV with the control group according to age and sex.

Comments 8: When writing scientifically, it is recommended to alter instances where the pronoun "we" is employed in the text to maintain impartiality and professionalism. Conversely, choose for more appropriate wording such as "the study unveiled" or "the findings suggest." To enhance the manuscript's integrity, it is necessary to address these specific cases. The subsequent alterations are advised for the specified lines: - Line 336: "...and therefore provides strong support..."

The statement made in line 338 suggests that it may contribute to the progression of IgA vasculitis and initiate...

The activation of sRAGE takes place on endothelial cells.

According to line 341 of the text, it enhances the inflammatory response by facilitating...

The unambiguous manifestation of HMGB1's participation in the inflammatory response in IgAV is evident.

According to line 319 of the book, there are significant positive correlations that offer more evidence.

Furthermore, it is worth noting that there is a noticeable difference in blood concentrations between individuals diagnosed with IgAV and the control group.

Line 322 states that the onset of the sickness occurred after a period of six months of observation. These findings indicate that HMGB1...

In line 323, it is stated that the high levels of IgAV have a special effect on its growth.

In line 324, the user mentions the early stage of disease manifestation, which is followed by an expected recovery.

Response 8: Thank you for suggestions on how to improve the above sentences and text of the manuscript. We have modified the most of them, it is marked with Track changes. We performed English language edition.

4. Response to Comments on the Quality of English Language

Point 1: Extensive English Revision is required.

To maintain objectivity and professionalism in scientific writing, it is recommended to revise instances where the term "we" is used in the manuscript. Instead, opt for more appropriate language such as "the study found" or "the results suggest." Addressing these instances will enhance the integrity of the manuscript. Here are the suggested revisions for the identified lines:

- Line 336: "...and therefore provides strong support…"

- Line 338: "...promoting IgA vasculitis and inducing..."

- Line 340: "...on endothelial cells, sRAGE activates…"

- Line 341: "...enhances the inflammatory response through promotion..."

- Line 318: "...role of HMGB1 in the inflammatory response in IgAV is evident..."

- Line 319: "...furthermore supported by significant positive correlations..."

- Line 321: "...In addition to the observed difference between IgAV patients and controls, serum concentrations..."

- Line 322: "...the onset of the disease and after a six-month follow-up interval. This may imply that HMGB1..."

- Line 323: "...has a certain impact on the development of IgAV itself, given its higher concentrations..."

- Line 324: "...the acute phase when disease appears, followed by an expected recovery…"

Addressing these recommendations comprehensively will significantly improve the manuscript's quality and impact, making it more suitable for publication and contributing meaningfully to the field of IgA vasculitis research.

Response 1: We have performed the English language editing as suggested.

5. Additional clarifications

Once again, we thank Reviewer 1 for taking time to revise our manuscript and for comments on how to improve it. We hope that our answers, as well as the limitations that exist in our study, will be overall acceptable.

Reviewer 2 Report

Comments and Suggestions for Authors

This is a prospective study of 86 patients with IgA vasculitis (IgAV) to evaluate four parameters (galactose-deficient immunoglobulin A1, Gd-IgA1; high mobility group box 1, HMGB1; receptor for advanced glycation end-product, RAGE; protocadherin 1, PCDH1) for biomarkers of disease diagnosis and activity. The authors demonstrated that serum HMGB1, RAGE and PCDH1 levels and urinary Gd-IgA1, HMGB1, RAGE and PCDH1 levels were significantly elevated in the IgAV group than those in controls. Moreover, serum HMGB1 and RAGE levels at onset were significantly elevated compared to those at remission in the IgAV group. Therefore, authors concluded that these parameters were useful as biomarkers of disease activity in IgAV.

The presented study was well performed, and the manuscript is described in a reasonable manner. However, the analysis methods were inadequate, in my opinion. To evaluate the usefulness of each parameter, it is necessary to calculate sensitivity and specificity using the receiver operating characteristic (ROC) curve analysis, in my opinion.

Moreover, I have several concerns as follows:

1.       Laboratory findings (Table 2) and four parameters (Table 3) were compared between two groups: the IgAV at onset group and control, the IgAV at follow-up group and control, and the IgAV at onset and follow-up groups, but those should be compared between three groups using ANOVA and post-hoc tests.

2.       Similar to above, when analyzing four indicators based on the presence or absence of nephritis (Table 4), the analysis should be performed across four groups rather than between each two groups.

3.       I think the footnote in Table 4 is incorrect, as it is the same as Table 3.

4.       Authors described that the logistic regression analysis was performed after model adjustment, but please provide all the adjusted indicators. Additionally, is it necessary to provide standard errors and Z values in Table 5?

Author Response

1. Summary

Thank you very much for taking the time to review this manuscript and provide your suggestions on how to improve it. Please find the detailed responses below and the corresponding revisions/corrections in track changes in the resubmitted files.

2. Questions for General Evaluation

Reviewer’s Evaluation

Response and Revisions

Does the introduction provide sufficient background and include all relevant references?

Yes

Are all the cited references relevant to the research?

Yes

Is the research design appropriate?

Yes

Are the methods adequately described?

Can be improved

Are the results clearly presented?

Must be improved

Are the conclusions supported by the results?

Must be improved

3. Point-by-point response to Comments and Suggestions for Authors

Comments 1: This is a prospective study of 86 patients with IgA vasculitis (IgAV) to evaluate four parameters (galactose-deficient immunoglobulin A1, Gd-IgA1; high mobility group box 1, HMGB1; receptor for advanced glycation end-product, RAGE; protocadherin 1, PCDH1) for biomarkers of disease diagnosis and activity. The authors demonstrated that serum HMGB1, RAGE and PCDH1 levels and urinary Gd-IgA1, HMGB1, RAGE and PCDH1 levels were significantly elevated in the IgAV group than those in controls. Moreover, serum HMGB1 and RAGE levels at onset were significantly elevated compared to those at remission in the IgAV group. Therefore, authors concluded that these parameters were useful as biomarkers of disease activity in IgAV. The presented study was well performed, and the manuscript is described in a reasonable manner. However, the analysis methods were inadequate, in my opinion. To evaluate the usefulness of each parameter, it is necessary to calculate sensitivity and specificity using the receiver operating characteristic (ROC) curve analysis, in my opinion.

Response 1: Thank you for pointing this out and finding it worth of displaying. We included ROC curve analysis in our revised manuscript and the results are presented in Table 5.

We also added and described ROC analysis in the Materials and Methods in the section related to statistical analysis.

Comments 2: Laboratory findings (Table 2) and four parameters (Table 3) were compared between two groups: the IgAV at onset group and control, the IgAV at follow-up group and control, and the IgAV at onset and follow-up groups, but those should be compared between three groups using ANOVA and post-hoc tests.

Response 2: You probably refer to the type of ANOVA called a "mixed-design analysis of variance (mixed-design ANOVA)," which is used to test for differences between two or more independent groups and allows repeated measures of the study participants. Mixed-design ANOVA would indeed be a very good fit for this type of data presentation. However, before the ANOVA can be approached, the criteria for the assumptions of linearity, homogeneity of variance, and sphericity must be met. Some of the variables within the data show evidence of non-normal data distribution (Shapiro-Wilk test, p<0,001) alongside skewness and kurtosis coefficients above 2.0 with p<0.001. Levene’s test also showed unequal variances between groups of patients and controls with HMGB and PCHD1 variables. Applying a test that assumes the normal data distribution of data would increase the chances of a false positive result (Type I error). To avoid this, the analysis between groups of unpaired data was done with an independent samples T-test in the case of normal data distribution and equal variances. The nonparametric Mann-Whitney U-test was selected in cases of non-normal data distribution because of its insensitivity to outliers and extremes. The paired samples were analyzed with a non-parametric Wilcoxon signed rank test. This type of results presentation is also more simplified and approachable. Additionally, analyzing some groups (such as IgAV control group and IgAV follow-up group) would not be plausible. 

Comments 3: Similar to above, when analyzing four indicators based on the presence or absence of nephritis (Table 4), the analysis should be performed across four groups rather than between each two groups.

Response 3: We wanted to examine whether there is a difference between patients with IgAV who have nephritis and those without at the onset of the disease and during the follow-up period. We didn’t consider it necessary to compare all four groups mutually.

Comments 4: I think the footnote in Table 4 is incorrect, as it is the same as Table 3.

Response 4: Thank you for the remark, we corrected the footnote.

Comments 5: Authors described that the logistic regression analysis was performed after model adjustment, but please provide all the adjusted indicators. Additionally, is it necessary to provide standard errors and Z values in Table 5?

Response 5: Univariate logistic regression was performed after model adjustments of potentially confounding variables (age, sex). We considered it important to present all data and coefficients of the analysis, and for this reason we included all the parameters of the logistic regression.

4. Response to Comments on the Quality of English Language

Point 1:

Response 1: (in red)

5. Additional clarifications

Once again, we thank Reviewer 2 for her/his time, kind comments of our manuscript and expertise in regard to the review of our manuscript. Thank you for your suggestions on how to improve our manuscript. We hope that our answers, as well as the limitations that exist in our study, will be overall acceptable.

Round 2

Reviewer 1 Report

Comments and Suggestions for Authors

1. The manuscript effectively compares serum and urine concentrations of Gd-IgA1, HMGB1, RAGE, and PCDH1 between IgAV patients and controls.
2. Consider providing additional context or interpretation of the findings to enhance understanding, particularly regarding the clinical significance of altered levels of these molecules in IgAV pathogenesis.
3. Ensure clear labeling and interpretation of figures, especially Figure 1. Additional explanation in the caption or main text could aid comprehension by highlighting key trends or significant differences observed in the data.
4. The manuscript provides a detailed overview of patient characteristics, treatment modalities, and outcomes.
5. Clarity and conciseness could be improved in presenting laboratory parameters and their comparisons.
6. Figures need clearer labeling and interpretation, especially Figure 1.
7. Minor revisions for language clarity and organization could enhance readability.

Author Response

1. Summary

Thank you very much for taking your time to review our manuscript and for suggestions and thoughtful comments how to further improve it. Please find the detailed responses below and the corresponding revisions/corrections highlighted/in track changes in the re-submitted files.

2. Questions for General Evaluation

Reviewer’s Evaluation

Response and Revisions

Does the introduction provide sufficient background and include all relevant references?

Can be improved

Are all the cited references relevant to the research?

Yes

Is the research design appropriate?

Can be improved

Are the methods adequately described?

Yes

Are the results clearly presented?

Yes

Are the conclusions supported by the results?

Yes

3. Point-by-point response to Comments and Suggestions for Authors

Comments 1: The manuscript effectively compares serum and urine concentrations of Gd-IgA1, HMGB1, RAGE, and PCDH1 between IgAV patients and controls.

Response 1: Thank you for your kind comment.

Comments 2: Consider providing additional context or interpretation of the findings to enhance understanding, particularly regarding the clinical significance of altered levels of these molecules in IgAV pathogenesis.

Response 2: We have added additional explanation regarding ROC analysis. Other results (logistic regression etc.) are explained in previous revision of the manuscript.

In the discussion we have explained what is potential clinical significance of the results (i.e. Since a certain number of patients will develop nephritis within clinical course of IgAV, measuring the concentration of HMGB1 in the urine may be a useful tool in evaluating IgAV patients with nephritis and in making therapeutic approaches. (…) Similar to serum HMGB1, the serum concentrations of RAGE differed between IgAV patients at baseline and after the follow-up interval. This may indicate residual low inflammatory activity or tissue damage despite clinical remission, suggesting that RAGE could have potential in disease activity evaluation and even as a therapeutic target.)

We explained the limitations of the study, which mostly refer to a relatively small sample, therefore a causal interference is restricted. We also emphasized in the conclusion that for any potential recommendations and application in clinical practice, we need further studies: (…While the overall results of our study should be interpreted cautiously, urinary HMGB1 particularly highlighted its potential. Since identifying a useful surrogate biomarker for nephritis in IgAV remains challenging, further larger scale studies are desirable and necessary to fully address this issue).

Comments 3: Ensure clear labeling and interpretation of figures, especially Figure 1. Additional explanation in the caption or main text could aid comprehension by highlighting key trends or significant differences observed in the data.

Response 3: Thank you for pointing this out. We have accordingly changed Figure 1. and we hope that in this form it will be easier for comprehension and acceptable for the overall content.

Comments 4: The manuscript provides a detailed overview of patient characteristics, treatment modalities, and outcomes.

Response 4: Thank you for your comment, we tried to include all relevant clinical and other characteristics which are generally considered important in patients with IgA vasculitis.

Comments 5: Clarity and conciseness could be improved in presenting laboratory parameters and their comparisons.

Response 5: Thank you for pointing this out. We have tried to highlight the results as clearly as possible with tabular presentations and emphasizing statistically significant values (Table 2). Please note that we analyzed laboratory parameters between patients at the onset of the disease and after six months follow up interval, and in control subjects, and we could only present such a large amount of data in a table. We felt that it was much better to present all the data together than to present them separately in several tables, and we believe that this should not create difficulties for the readership in following the results.

Comments 6: Figures need clearer labeling and interpretation, especially Figure 1.

Response 6: We have accordingly modified and changed Figure 1. with an additional explanation of the text below the figure, and we believe that in this improved form it will be in overall acceptable.

Comments 7: Minor revisions for language clarity and organization could enhance readability.

Response 7: Thank you for pointing this out. We corrected misspellings and once again carefully went through manuscript in spelling and written form for overall intelectual content.

4. Response to Comments on the Quality of English Language

Point 1:

Response 1: (in red)

5. Additional clarifications

Once again, we thank Reviewer 1 for taking time to revise our manuscript and for expertise in regard to the review of our manuscript. Thank you for all suggestions and positive comments on how to improve it. We hope that our answers, as well as the limitations that exist in our study, will be overall acceptable.

Reviewer 2 Report

Comments and Suggestions for Authors

This is a prospective study of 86 patients with IgA vasculitis (IgAV) to evaluate four parameters (galactose-deficient immunoglobulin A1, Gd-IgA1; high mobility group box 1, HMGB1; receptor for advanced glycation end-product, RAGE; protocadherin 1, PCDH1) for biomarkers of disease diagnosis and activity.

The presented study was well performed and the revised manuscript is described in a reasonable manner. Authors had also responded to my all comments.

Author Response

1. Summary

Thank you very much for taking the time to review this manuscript and provide your suggestions on how to improve it.

2. Questions for General Evaluation

Reviewer’s Evaluation

Response and Revisions

Does the introduction provide sufficient background and include all relevant references?

Yes

Are all the cited references relevant to the research?

Yes

Is the research design appropriate?

Yes

Are the methods adequately described?

Yes

Are the results clearly presented?

Yes

Are the conclusions supported by the results?

Yes

3. Point-by-point response to Comments and Suggestions for Authors

Comments 1: This is a prospective study of 86 patients with IgA vasculitis (IgAV) to evaluate four parameters (galactose-deficient immunoglobulin A1, Gd-IgA1; high mobility group box 1, HMGB1; receptor for advanced glycation end-product, RAGE; protocadherin 1, PCDH1) for biomarkers of disease diagnosis and activity. The presented study was well performed and the revised manuscript is described in a reasonable manner. Authors had also responded to my all comments.

Response 1: We would like to thank reviewer 2 for taking the time to review our manuscript and and for expertise in reviewing our manuscript. Thank you for all the suggestions and positive comments that have constructively improved our manuscript.